# Do Patients Benefit from Micronutrient Supplementation following Pancreatico-Duodenectomy?

**DOI:** 10.3390/nu15122804

**Published:** 2023-06-19

**Authors:** Mary E. Phillips, Kathryn H. Hart, Adam E. Frampton, M. Denise Robertson

**Affiliations:** 1Department of Nutrition and Dietetics, Royal Surrey Hospital, Guildford GU2 7XX, UK; 2Faculty of Health and Medical Sciences, University of Surrey, Guildford GU2 7NX, UK; k.hart@surrey.ac.uk (K.H.H.); m.robertson@surrey.ac.uk (M.D.R.); 3Oncology Section, Surrey Cancer Research Institute, Department of Clinical and Experimental Medicine, FHMS, University of Surrey, The Leggett Building, Daphne Jackson Road, Guildford GU2 7WG, UK; adam.frampton@surrey.ac.uk; 4Department of HPB Surgery, Royal Surrey Hospital, Guildford GU2 7XX, UK

**Keywords:** micronutrients, pancreatico-duodenectomy, deficiency, supplementation

## Abstract

Pancreatico-duodenectomy (PD) includes resection of the duodenum and use of the proximal jejunum in a blind loop, thus reducing the absorptive capacity for vitamins and minerals. Several studies have analysed the frequency of micronutrient deficiencies, but there is a paucity of data on those taking routine supplements. A retrospective review of medical notes was undertaken on 548 patients under long-term follow-up following PD in a tertiary hepato-pancreatico-biliary centre. Data were available on 205 patients from 1–14 years following PD, and deficiencies were identified as follows: vitamin A (3%), vitamin D (46%), vitamin E (2%), iron (42%), iron-deficiency anaemia (21%), selenium (3%), magnesium (6%), copper (1%), and zinc (44%). Elevated parathyroid hormone was present in 11% of cases. There was no significant difference over time (*p* > 0.05). Routine supplementation with a vitamin and mineral supplement did appear to reduce the incidence of biochemical deficiency in vitamin A, vitamin E, and selenium compared to published data. However, iron, vitamin D, and zinc deficiencies were prevalent despite supplementation and require surveillance.

## 1. Introduction

Pancreatico-duodenectomy (PD) is a complex surgical procedure and is associated with malnutrition, pancreatic exocrine insufficiency (PEI), and sarcopenia [1,2]. 

Due to short-term survival, outcome measures have focussed on immediate post-operative complications [3,4] and quality of life [5]. However, as survival increases, there is a developing need for more long-term support for these patients. 

The majority of studies analysing outcomes from this operation focus on patients with pancreatic ductal adenocarcinoma (PDAC) [6]. Most cases of PDAC are inoperable at the time of diagnosis, with less than 20% of cases operable at diagnosis, and have high recurrence rates, with a median survival less than 2 years [7]. However, for those patients operable at diagnosis and able to tolerate adjuvant chemotherapy, the prognosis is much better, with clinical trials reporting 5-year survival rates of 28–30% [8,9]. 

Whilst PDAC remains the most likely reason for undertaking PD, this procedure is also carried out for cancers of the duodenum, distal common bile duct, and ampulla. PD is also carried out for benign diseases, predominantly chronic pancreatitis (CP) [10], and pre-malignant diseases, such as intrapapillary mucinous neoplasm (IPMN) [11] and familial adenomatous polyposis (FAP) [12]. Less-aggressive tumours, such as pancreatic neuroendocrine tumours (pNET), may also occur in the pancreas and are also treated with PD [13,14], and in some instances, PD may be carried out for benign diseases [15]. 

With a combination of benign and pre-malignant diseases and with 5-year survival rates for malignant diseases as high as 30%, there is clear justification for work exploring the long-term management of these patients. 

Many vitamins, minerals, and trace elements are absorbed in the duodenum and proximal jejunum, the former of which is removed during the PD resection and the latter used in the reconstruction and therefore out of continuity. Furthermore, this surgery is associated with a reduction in gastric acid secretion due to either resection of the distal stomach or, in pylorus-preserving PD, long-term use of proton-pump inhibitors (PPI), as well as fat malabsorption due to pancreatic exocrine insufficiency (PEI).

National guidelines in the United Kingdom recommend the nutritional monitoring of patients following treatment for pancreatic cancer [16], and biochemical vitamin and mineral testing is recommended in national and European guidelines as part of the long-term follow-up of patients with chronic pancreatitis [17,18]. 

Long-term follow up of patients following PD has been in place for over 15 years in our institution, and during this time, patients were routinely supplemented with a multivitamin and mineral and a calcium and vitamin D supplement. This practice commenced following the diagnosis of micronutrient deficiencies in our long-term survivors [19]. 

Whilst micronutrient deficiencies are reported as prevalent in the literature for unsupplemented patients, we aimed to investigate the incidence of micronutrient deficiency in those receiving routine supplementation. 

## 2. Materials and Methods

A retrospective review of micronutrient levels was undertaken in all patients under follow-up after PD in a tertiary pancreatic resection centre. This study was approved by the clinical audit department (registration number 961).

### 2.1. Data Selection

Blood test results were retrieved from paper and electronic records for all patients logged on a prospectively maintained database of those under follow-up for at least 2 years with the dietitians following PD from 2005 to 2021. Data collection stopped in 2021 due to a change in laboratory assays, resulting in data sets that were not comparable with historic data. 

Biochemical testing consisted of measuring levels of vitamins A, D, and E, selenium, zinc, copper, magnesium, anaemia-screening substances (including ferritin, iron, transferrin, haemoglobin, vitamin B12, and folate), renal function, liver function, C-reactive protein (CRP), and parathyroid hormone (PTH). 

Patient records were assessed for any comments regarding compliance with supplements. 

### 2.2. Definitions

Iron deficiency was determined if serum ferritin levels were below 30 Ug/L or transferrin saturations below 15%. Iron-deficiency anaemia was defined as iron deficiency in combination with a low haemoglobin (<115 g/L for females and <130 g/L for males). Reference ranges were taken from the laboratory that analysed the blood samples and are summarised in Table 1.

### 2.3. Confounding Variables 

CRP was used as a marker of inflammation, and blood tests with a CRP exceeding 10 mmol/L [20] were excluded. 

All blood tests were analysed through the same laboratory, and local protocols included protection from ultraviolet light and timely processing for vitamin A and E assays. 

Blood tests were not carried out on patients with disease recurrence, undergoing palliative care, or whilst acutely unwell, which is routine clinical practice within our unit. 

Malabsorption due to PEI may reduce the uptake of micronutrients. Pancreatic enzyme replacement therapy (PERT) is prescribed regularly in our centre and optimised within the first few weeks of surgery. All patients were taking PERT with a minimum dose of 50,000 units of lipase with meals and 25,000 units with snacks. All patients were seen annually by the specialist dietitian for the purpose of optimising PERT and glycaemic control. Data on dose escalation are outside of the scope of this study. 

All patients were prescribed a proton-pump inhibitor to reduce the risk of ulceration at the gastro-jejunostomy. 

### 2.4. Micronutrient Supplementation

All patients were prescribed and reported taking a single vitamin and mineral supplement, either Sanatogen A–Z Complete^®^ (Bayer, Reading, UK) or Forceval^®^ (Alliance Pharmaceuticals, Chippenham, UK) and a combined calcium and vitamin D supplement, either Adcal D3^®^ (Kyowa Kirin Ltd., Galashiels, UK), Calcichew D3^®^ (Neon Healthcare, Hertford, UK), or Accrete D3^®^ (Internis Pharmaceuticals, Huddersfield, UK), twice daily, the contents of which are summarised in Table 2 and compared to the Reference Nutrient Intake (RNI), which are the quantities estimated to meet the nutritional requirements of 97.5% of the United Kingdom (UK) population. The products used were determined by local prescriptive recommendations. 

Patients with mild deficiencies were treated with double doses of their multivitamin and mineral, and those with more severe deficiencies were treated with singular micronutrient supplements. Solvazinc^®^ (45 mg of elemental zinc three times a day for three months), Selenase^®^ (200 µg once daily for three months), ferrous fumerate (210 mg twice daily for three months), and high-strength vitamin D supplements (providing a total of 300,000 units over 6–10 weeks) were prescribed as needed. Supplementation was re-evaluated at three months. If biochemical levels returned to normal, additional supplements were discontinued and the patients re-evaluated at their next annual reviews. If there was some improvement but results had not yet normalised, another 3 months of supplementation was prescribed. If there was no improvement at three months, intramuscular (vitamin D only) or intravenous supplementation was provided. These additional biochemical results were not included in this study, as they were undertaken at different time frames and in different laboratories.

### 2.5. Statistical Analysis

Descriptive statistics were performed to identify the incidence of deficiency in each year following PD. Data were assessed for normality using the Shapiro–Wilk test and non-parametric tests carried out using the Kruskal–Wallis one-way ANOVA. Data were analysed in SPSS, version 26 (IBM, Armonk, NY, USA), and significance was assumed at *p* < 0.05.

## 3. Results

Data were available on 548 patients, of which 343 (63%) were excluded, as they had fewer than 2 years of follow-up, of which 129 patients had early recurrence (disease recurrence within 2 years and passed away or were having palliative treatment at 24 months from surgery). One surgeon discharged patients to local follow-up after 6 months (*n* = 105), and some patients declined follow-up at the tertiary centre, failed to attend, or were followed up in the private sector (*n* = 33, 12, and 49, respectively). Fifteen patients were lost to follow-up (Figure 1). Therefore, 205 patients were included in this study. Five individual data sets were excluded in the analysis, as CRP was greater than 10 mg/L.

Of the 205 patients followed up over 14 years, 104 (50.7%) were men and the mean age was 61.8 (SD 11.8). One hundred sixty-five (80.5%) underwent a pylorus-preserving pancreatico-duodenectomy, thirty-two a full Whipple, and eight a total pancreatectomy. Of all 205 patients, 5 had additional resections (2× right hepatectomy, 1× vascular resection, 1× hemicolectomy, and 1× nephrectomy). The most common pathological diagnosis was pancreatic ductal adenocarcinoma (PDAC), followed by pre-malignant conditions (intrapapillary mucinous neoplasm (IPMN), familial adenomatous polyposis (FAP), and tubular villous adenoma), with the benign and pre-malignant disease pathologies more prevalent in those long-term survivors (Figure 2). 

Most patients had more than one vitamin and mineral screen, and all data sets were included. Between 93 and 384 results were available for each micronutrient, with data being most available for vitamin D (*n* = 384), vitamin B12 (*n* = 383) vitamin A (*n* = 382), vitamin E (*n* = 371), zinc (*n* = 264), and selenium (*n* = 361). Data were least available for copper (*n* = 93), parathyroid hormone (*n* = 237), and magnesium (*n* = 282). Diabetes (defined by an HbA1c > 44 mmol/mol) was present in 42% of all blood tests (*n* = 380) (Appendix A).

Fat-soluble-vitamin biochemical deficiencies were identified in 3% of cases for vitamin A and 2% vitamin E, and 46% of results demonstrated vitamin D deficiency. A total of 11% of patients had elevated parathyroid hormone, suggesting disordered calcium homeostasis, and these were all associated with vitamin D deficiency (Appendix A). One patient required intravenous fat-soluble vitamins over a course of 3 months to correct deficiencies; all others responded to high-dose oral supplements. 

Trace-element deficiencies were 1% copper, 3% selenium, 6% magnesium, and 44% zinc. Iron-deficiency anaemia was present in 21% of cases, while 42% were iron-deficient. Vitamin B12 deficiency was only found in three cases (1%) (Appendix A). All deficiencies responded to oral supplementation, aside from one patient who was treated with intravenous iron after oral iron did not result in an improvement. 

The most prevalent deficiencies were zinc (44%), vitamin D (46%), and iron—assessed as ferritin (42%) (Appendix A). These did not change significantly with time from surgery (*p* = 0.636, *p* = 0.774, and *p* = 0.257, respectively) (Figure 3). This is despite each individual deficiency being corrected and reflects a new diagnosis of deficiency in those where more than one result was available. There were not sufficient data to carry out repeated measurement analysis on individual patients.

Folate levels appeared to increase with time, but this did not reach significance (*p* = 0.136). 

A total of 44% of folate results were above the upper reference range. Only one case of folate deficiency was identified (Figure 4).

Compliance was only commented on four times within the medical notes and therefore could not be assessed. 

## 4. Discussion

We evaluated the benefits of routine supplementation in patients after PD. Whilst there are some data exploring the incidence of micronutrient deficiencies in this cohort, these are predominantly limited by small sample sizes and wide variations in time from surgery [22,23], which will introduce confounding variables due to concurrent chemotherapy and varying body stores (Table 3).

Disease pathology is also important, and guidelines are in place for those with pancreatic cancer and pancreatitis [16,17]. This study demonstrates that these guidelines on follow-up should extend to those with pre-malignant diseases and other malignant diseases (Figure 2).

The first study detailing micronutrient status was conducted in 1991 and explored the outcomes of total pancreatectomy in 45 consecutive patients who underwent surgery between 1978 and 1988 and 4 patients who had surgery from 1972-6 [24]. A total of 47 patients had a distal gastrectomy as part of this resection. This study highlighted a comprehensive follow-up protocol with annual dietetic assessments, high-dose vitamin D supplementation, 1000 mg of calcium, and a double dose of a multivitamin. Eleven patients had bone density assessments after 5 years, demonstrating a 17.6% reduction in bone mass in the first 5 years [24]. Despite this intensive regimen, 20% of patients were biochemically deficient in vitamin D, suggesting that intake is not the only factor determining biochemical deficiency. This study also quantified the energy intake of patients at 56 ± 1 kcal/kg in those who were weight-stable beyond 18 months from surgery, despite pancreatic enzyme replacement therapy. The PERT dose was documented at 3–5 capsules with meals and 2–3 with snacks. However, the strength of the PERT was not specified in the study’s publication but detailed as a minimum of 20,000 units of lipase per main meal within the discussion, suggesting these were likely to be capsules containing no more than 10,000 units of lipase. Furthermore, faecal fat losses of 17%, confirming ongoing steatorrhoea, were reported [24]; thus it is likely that, whilst considered a high dose at the time, the dose of PERT provided was too low and was certainly well below the dose currently recommended [25]. However, this study remains the most comprehensive study exploring long-term nutritional outcomes identified in the literature.

### 4.1. Trace-Element Deficiency

#### 4.1.1. Zinc

Zinc is absorbed primarily in the jejunum and is secreted into bile and reabsorbed through the enterohepatic circulation [26]. Deficiency may be due to reduced absorptive capacity and increased losses due to excess biliary losses. Bile salt malabsorption has been documented in up to 75% of patients after pancreatic surgery [27]. Yu et al. documented a high level of zinc deficiency, but 18.7% of the cohort reported symptoms of significant steatorrhoea, and the treatment dose of PERT was low, at 20,000 with meals [28]; thus, ongoing malabsorption may have contributed to the high incidence of zinc deficiency. Zinc deficiency was common in our cohort, and this was consistent over 13 years of follow-up, despite supplementation in excess of the RNI. High-dose oral zinc was prescribed to treat deficiencies, and biochemical assessment was repeated 3 months after supplementation to ensure this was effective. Intravenous supplementation was not required in any case. Levels of deficiency in our cohort were consistent with other studies [22,29,30], demonstrating that supplementation with a multivitamin does not reduce the incidence of zinc deficiency, although levels improve with high-dose oral supplementation. Importantly, regardless of correction, the incidence of zinc deficiency remained consistent over 10 years, confirming the need for long-term follow-up (Figure 3).

#### 4.1.2. Selenium

Selenium is readily absorbed in the proximal intestine, and deficiencies are seen in those who have undergone upper-gastrointestinal surgery [31]. However, most documented cases are associated with dietary deficiency [26] and corrected with oral supplementation. The incidence of selenium deficiency was much lower in our cohort (3%) compared to the reported incidences of 24–47% in other studies [23,30,32]. However, the incidence was higher in the first 4 years (but never exceeded 11%). Given the ready absorption, the low incidence of deficiency may be due to the provision of 50 µg in the multivitamin supplement. Whilst this does not meet the RNI for selenium (75 µg), it may be sufficient, alongside a normal diet to prevent the deficiencies seen in non-supplemented cohorts. 

#### 4.1.3. Copper

European guidelines recommend the measurement of copper levels in patients who have had surgery that excludes the duodenum [26], with cases reported following bariatric surgical procedures, which have a similar anatomical impact as PD. The same guidelines also recommend oral supplementation in chronic conditions, and given the low levels of copper deficiency in our cohort, oral supplementation seems effective, and biochemical surveillance may not be necessary.

#### 4.1.4. Magnesium

Magnesium is considered to be a helpful biochemical marker for malabsorption [33]. In our cohort, magnesium deficiency was present in 6% of cases, but it was only mild in all cases (with all results greater than 0.6 mmol/L). Singular supplementation was not required, and patients responded to increased doses of PERT and a double dose of vitamin and mineral supplement. Active management of PEI within our cohort may have reduced the incidence of magnesium deficiency, and monitoring of serum levels may only be necessary in those with ongoing malabsorption. 

### 4.2. Assessment of Anaemia

All markers of iron stores are affected by inflammation, and in this setting, iron deficiency was prevalent and a fifth of patients were anaemic. This may be a result of reduced uptake of iron due to duodenal resection [26]. Dietary intake was likely to be sufficient, as vitamin and mineral supplements contain 150% of the Reference Nutrient Intake (RNI), but despite this, additional iron supplementation was needed. Although there was no significant difference in ferritin levels over 10 years (*p* = 0.257, Figure 3), anaemia was most prevalent in the early years, with very few cases after 5 years. This may reflect adaptation or previous supplementation. 

As part of the anaemia screen, vitamin B12 and folate were assessed, and whilst only one case of low folate levels was identified, folate was elevated in 44% of cases. Folic acid supplementation provided between 100 and 200% of the RNI, which may account for the high levels. Other causes of high folate include vitamin B12 deficiency, but as this had a prevalence of less than 1%, this is unlikely to be a factor in the current study. Other studies have found that increased folate, hypothesized to be produced by bacterial fermentation, has been identified in patients with small-intestinal bacterial overgrowth (SIBO) [34]. SIBO has been documented in patients with chronic pancreatitis and after pancreatic surgery [35], as the presence of a blind loop of bowel is widely recognised as a significant risk factor [34]. A study of 545 pancreatic surgical patients found ongoing bowel symptoms in 67% of patients, but only 20% were referred for assessment for SIBO, with a prevalence in this study of 5.5% (but over 25% of those were actually assessed) [35]. A complicating factor in the diagnosis of SIBO may well be the symptoms of diarrhoea, flatulence, abdominal bloating, and discomfort [34], which mimic those of PEI, which occurs in nearly all patients following PD [1]. Elevated folate levels in patients with ongoing bowel symptoms could be a potential trigger for assessment of SIBO.

### 4.3. Fat-Soluble Vitamin Deficiency

Body stores may explain the lack of vitamin A deficiency in the first few years. Tabriz et al. explored vitamin and mineral levels in the first year post-PD and concluded that, as there was no vitamin A deficiency but significant vitamin D deficiency, that fat-soluble-vitamin malabsorption was not to blame [30]. However, vitamin A stores are thought to be sufficient to prevent deficiency for many months, even in those with zero vitamin A intake [21], which may explain the discrepancies between the short-term and long-term deficiencies. PERT supplementation was documented at 2000 units of lipase per gram of fat in this study [30], equivalent to 40,000 units for a 20 g fat meal. In our cohort, 10 instances of biochemical vitamin A deficiency occurred, but all 10 cases were in the same 2 patients, both of whom had a long history of chronic pancreatitis and malabsorption prior to surgery. In both cases, the deficiency recurred despite treatment, and one patient demonstrated clinical signs of vitamin A night blindness and was treated with intravenous vitamin A supplementation. 

Vitamin D deficiency was common and consistent across all year groups at 46%, and there was no difference over time (Figure 3). This is consistent with other studies and higher than the general population, where vitamin D deficiency has been reported at 24% [36]. This was despite supplementation (15–20 µg) in excess of the UK recommendations for winter months (10 µg); thus, it is likely there is poor uptake of dietary supplements. 

Vitamin E deficiency was uncommon but more prevalent after 5 years, which may reflect loss of body stores, and serum vitamin E does not drop until body stores are completely depleted [21]. One patient, who previously had vitamin A deficiency night blindness, presented with hemi-paresis, and after neurological conditions were excluded, serum vitamin E was recorded at <10 nmol/L. Following intravenous supplementation, his neurological symptoms resolved, and the patient was maintained on high-dose vitamin A, D, and E supplements. 

### 4.4. Bone Health

Tabriz et al. also demonstrated increasing levels of parathyroid hormone with time after PD, with the mean value at the top end of the reference range, with >25% of results classified as hyperparathyroid [30]. This supports the reduction in bone mineral density seen in other studies [24]. Vitamin D deficiency was only documented graphically in this study and was greater than 50% [30], which was similar to our incidence of 46%. However, only 11% of our cohort had elevated parathyroid hormone, which may reflect supplementation with calcium and vitamin D, but it is not possible to draw significant conclusions from these data. Further work should include bone density assessment. 

### 4.5. Limitations

#### 4.5.1. Deficiency within the General Population

It is important to consider that some micronutrient deficiencies occur in the general population and may be of dietary or environmental origin rather than a result of malabsorption. The National Diet and Nutrition Survey (NDNS) reported low intakes of zinc, iron, calcium, magnesium, and selenium in the general population, but biochemical testing was not available [37].

Armstrong et al. compared serum micronutrients in patients following PD with those of their spouses to try and correct for lifestyle and dietary intake, assuming that people who live together follow similar meal patterns and lifestyles [29]. They identified significantly lower levels of ferritin, selenium, vitamin D, and vitamin E (all *p* < 0.001) in those who had undergone PD compared to their spouses and furthermore found elevated parathyroid hormone in 30% of patients, compared to none of the spouses, suggesting elevated risk of osteoporosis due to bone depletion of calcium [29]. An earlier study exploring dietary adequacy in a small cohort of 10 patients found low dietary intakes of vitamins A, D, and E, zinc, iron, and selenium in patients following PD, but this was not compared with a control group [23].

#### 4.5.2. Inflammation

Inflammation was adjusted for by the exclusion of results with a CRP > 10 mg/L. In comparison, only one previous study identified the differences in results when corrected for inflammation and found increased incidences of selenium, iron, and zinc deficiencies when those with high CRP levels were included [32] (Table 3), which may reflect the impact of inflammation. Armstrong et al. excluded patients with high C-reactive protein (CRP) levels [29], whereas Murphy et al. identified high CRP levels in 42% of the patients screened [32]. Whilst the latter study explored pre-operative patients, it did not identify those patients who had presented with jaundice/cholangitis, which may explain the number of patients with elevated inflammatory markers. Furthermore, hepatocyte damage from cholestasis, in addition to inflammation, may explain the high levels of ferritin observed [38]. Thus, CRP and liver function tests should always be measured alongside micronutrient status [26].

#### 4.5.3. Missing Data

Missing data were a significant limitation and occurred due to the retrospective nature of this study. There were missing data from 2020–2021 due to lack of non-urgent biochemical testing during the COVID-19 pandemic. Since the pandemic, the use of more telephone appointments further restricted access to phlebotomy. The incidence of missing data within this time frame resulted in insufficient data to carry out repeated measurement assessments. Thus it was not possible to see the change in biochemical levels over time for individual patients. 

#### 4.5.4. Compliance

It was not possible to accurately assess compliance due to the retrospective nature of the study. Whilst the authors looked for evidence of compliance in medical documentation, this was rarely questioned beyond a medication history. Compliance with micronutrient supplements should be included in future prospective studies. 

#### 4.5.5. Ongoing Malabsorption

Micronutrient levels may be reduced if the patient has ongoing malabsorption. Regular dietetic review and adjustment of pancreatic enzyme replacement therapy is available within our institution, and no overt symptoms of uncontrolled malabsorption were noted when the medical notes were reviewed. Prospective work should include a more objective marker of malabsorption. 

#### 4.5.6. Accuracy of Biochemical Markers

Biochemical assessment of micronutrients has several limitations. In addition to the impact of body stores and inflammation, fat-soluble-vitamin levels deteriorate with exposure to UV light and over time. There are several relationships between individual micronutrients. For example, zinc plays a key role in the absorption and transport of vitamin A [39], iron metabolism is modulated by vitamin A [40], and iron is essential for vitamin D synthesis [41]. 

Interestingly, although the long-term prescription of proton-pump inhibitors is associated with deficiencies in selenium and vitamin B12 [26], this was not observed in this study and may reflect the reduction in bicarbonate from the pancreas. 

Thus, interpretation of micronutrient deficiencies requires careful holistic consideration by an experienced clinician to ensure results are interpreted correctly. 

#### 4.5.7. Impact

Deficiencies in vitamin D, zinc, and selenium are associated with sarcopenia [26], which is associated with worse outcomes for pancreatic cancer [42]. Osteoporosis carries a high health care cost [43], but the clinical and financial impacts of other deficiencies are difficult to quantify and represent in clinical studies. Further work should explore the incidence of osteoporosis, clinical manifestations, and hospital admissions associated with micronutrient deficiencies.

**Table 3 nutrients-15-02804-t003:** Comparison of the incidences of micronutrient deficiencies in diet and biochemical analysis in patients before and after pancreatico-duodenectomy.

	Published Studies Reporting the Incidences of Micronutrient Deficiencies in Patients Undergoing Pancreatico-Duodenectomy							
	Dresler et al., 1991 [24]	Armstrong et al., 2002 [23]	Armstrong et al., 2007 [29]	Yu et al., 2011 [28]	Latenstein et al., 2021 [44]	Tabriz et al., 2021 [30]	Murphy et al., 2021 [32]	Murphy et al., 2021 [32]
Time from surgery (months)	6–117	Median 19 (6–46)	Median 23 (8–84)	>6	Median 13 (7–28)	12	0	0
Inflammation	NS	NS	Exclude	NS	NS	NS	Include	Exclude
Deficiency type	B	D	B	B	B	B	B	B
Supplements	Vitamin D and multivitamin	No	No	No	No	No	No	No
N	15	10	37	48	85	47	48	28
Vitamin A	27%	20%	NS	NS	21%	0	4%	0
Vitamin D	20%	100%	24%	NS	40%	>50%	58%	57%
Vitamin E	7%	20%	NS	NS	3%	NS	0	0
Zinc	7%	50%	NS	68%	4%	<25%	79%	83%
Iron		50%	NS	17.1%	34% F; 60% M	NS	33%	19%
Magnesium	27%	NS	NS	NS	NS	~50%	NS	NS
Selenium	NS	40%	56%	NS	NS	~25%	38%	24%

NS—not stated; B—biochemical; D—dietary; F—female; M—male.

## 5. Conclusions

Micronutrient deficiency was common in this population. Routine supplementation with a vitamin and mineral supplement did appear to reduce the incidence of biochemical deficiency in fat-soluble vitamins and selenium. However, iron, vitamin D, and zinc deficiencies were prevalent despite supplementation and require surveillance. This study confirms that life-long surveillance is required.

Vitamin A and vitamin E deficiencies only occurred early in the post-operative setting in patients who had significant malabsorption prior to surgery and do not require assessment until 5 years post-operatively in those whom malabsorption is controlled with PERT.

Based on these results, it may be possible to stratify biochemical testing in patients taking a vitamin and mineral supplement. For instance, annual surveillance of iron studies, vitamin D, zinc, and parathyroid hormone should continue, but surveillance of selenium, vitamin A, vitamin E, magnesium, copper, and B12 could be reserved for those with ongoing symptoms of malabsorption or if clinical symptoms suggest deficiency.

Further work should explore the use of elevated folate levels, in the presence of ongoing bowel symptoms, as a trigger for SIBO investigations. Prospective studies could include analysis of minerals not routinely analysed to ensure there are no other factors that should be included in clinical practice. Finally, it is important to include markers of inflammation in all micronutrient assessments.

## Figures and Tables

**Figure 1 nutrients-15-02804-f001:**
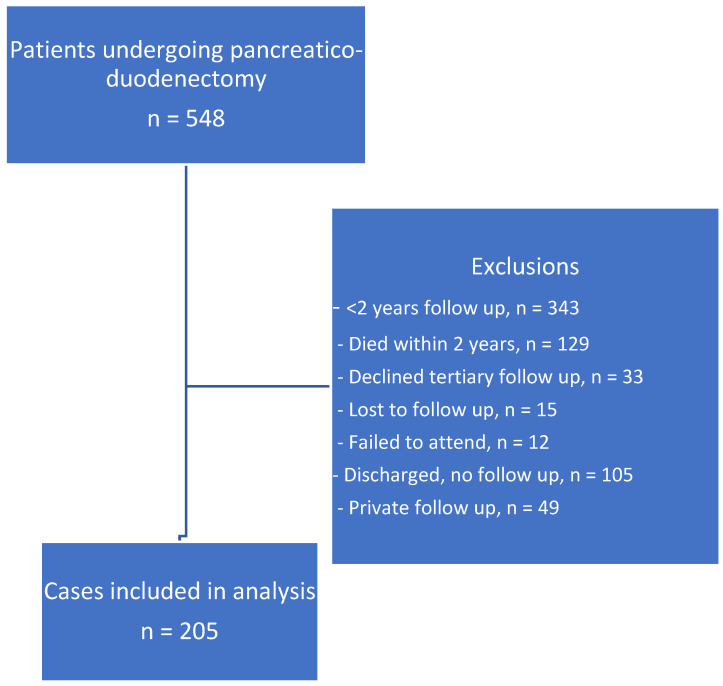
Data selection strategy.

**Figure 2 nutrients-15-02804-f002:**
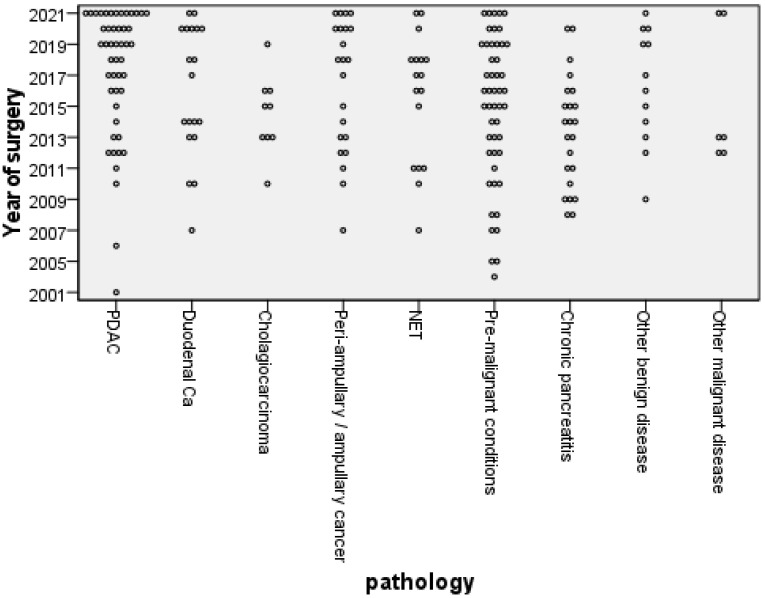
Distribution of pathological diagnosis by year of operation (PDAC, pancreatic ductal adenocarcinoma; Ca, Cancer; NET, neuroendocrine tumour).

**Figure 3 nutrients-15-02804-f003:**
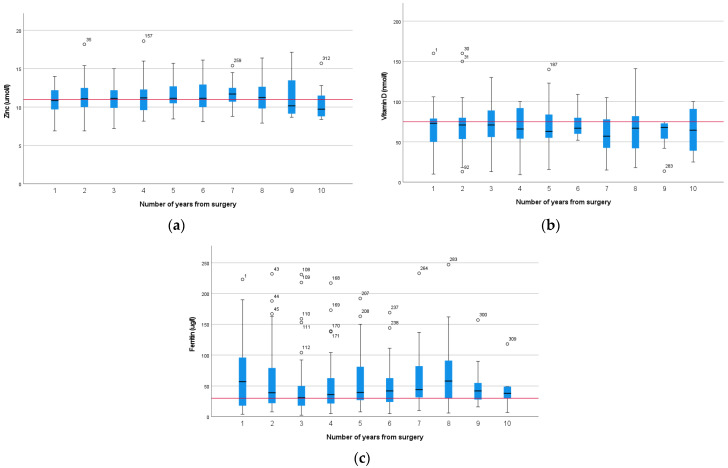
Box plots demonstrating change over time (years since surgery) for zinc (**a**), vitamin D (**b**), and ferritin (**c**) levels in patients (*n* = 364, 384, 338, respectively) following pancreatico-duodenectomy (The red line represents the lower reference range). Outliers are represented by hollow circles.

**Figure 4 nutrients-15-02804-f004:**
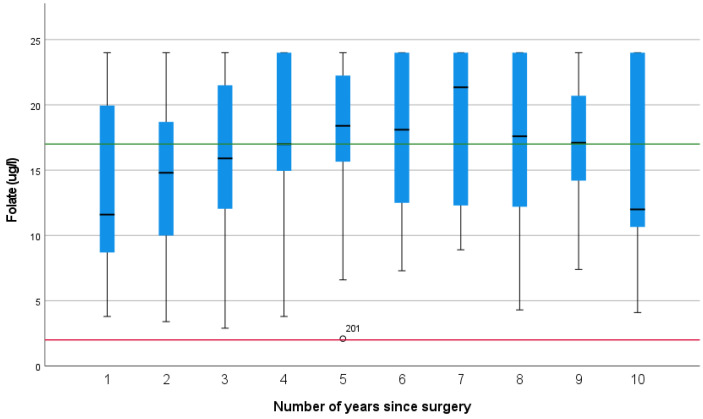
Folate levels over the first 10 years after pancreatico-duodenectomy (The green line represents the upper reference range, and the red line represents the lower reference range).

**Table 1 nutrients-15-02804-t001:** Vitamin and mineral reference ranges (Partnership Pathology, 2021).

Blood Test	Units	Reference Range
Vitamin A	µmol/L	0.99–3.35
Vitamin D	nmol/L	75–200
Vitamin E	µmol/L	9.5–41.5
Vitamin B12	ng/L	200–900
Folate	µg/L	2–17
Selenium	µmol/L	0.66–1.57
Zinc	µmol/L	11–24
Copper	µmol/L	10–22
Magnesium	Mmol/L	0.7–1
Ferritin	µg/L	30–250
Iron	µmol/L	9–30
Transferrin	g/L	2.5–3.8
Transferrin saturations	%	15–50
Haemoglobin—female	g/L	115–165
Haemoglobin—male	g/L	130–180
Parathyroid hormone	pmol/L	2–8.5
CRP	Mg/L	0–10

**Table 2 nutrients-15-02804-t002:** Composition of combined vitamin and mineral supplements compared to the Reference Nutrient Intake (RNI) for the UK.

Vitamin/Mineral	Forceval^®^	Sanatogen A–Z^®^	Adcal D3^®^ ×2	Calcichew D3^®^ ×2	Accrete D3^®^ ×2	UK RNI [21]
Vitamin A (µg)	750	906				700
Vitamin B1 (mg)	1.2	1.4				1.2
Vitamin B2 (mg)	1.6	1.6				1.3
Vitamin B3, nicotinamide (mg)	18	18				17
Vitamin B5, pantothenic acid (mg)	4	6				-
Vitamin B6 (mg)	2	2				1.4
Vitamin B7, biotin (µg)	100	150				30
Vitamin B9, folic acid (µg)	400	200				200
Vitamin B12 (µg)	3	1.3				21.5
Vitamin C (mg)	60	60				40
Vitamin D (µg)	10	5	10	10	10	-
Vitamin E (mg)	10	11				15
Vitamin K1 (µg)	-	20				-
Calcium (mg)	100	200	600	500	600	700
Iron (mg)	12	14				8.7
Copper (mg)	2	1				1.2
Magnesium (mg)	30	100				300
Zinc (mg)	15	15				9.5
Iodine (µg)	140	150				140
Manganese (mg)	2	1				-
Potassium (mg)	4	N/A				3500
Phosphorus (mg)	77	145				550
Selenium (µg)	50	55				75
Chromium (µg)	200	25				-
Molybdenum (µg)	250	N/A				-

RNI—Reference Nutrient Intake, mg—milligram, µg—microgram, N/A—not available.

## Data Availability

Data are contained within the Appendix A. The data presented in this study are available in Appendix A.

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
