# Peer review of "Do Patients Benefit from Micronutrient Supplementation following Pancreatico-Duodenectomy?"

_nutrients, 2023, doi:10.3390/nu15122804_

Round 1
Reviewer 1 Report
Fig.2: Please explain the need for this figure. I could not find discussion on this data.
Fig.3: Although most conclusions depend on Fig.3, there is no detailed explanation.
Table 3: Literature should be properly cited (please check the reference numbers).
Line 41: Ref. 26 gives no information on the zinc absorption mechanism.
Line 45: Ref.28 should be Ref.29 (Check the validity of references throughout the text).
Line 65: "high dose of zinc deficiency" should be checked.
Line 52-53: There appears to be no evidence in this study that high-dose oral zinc supplementation improves zinc levels.
Author Response
Thank you for your comments, I have answered the queries raised below with your original comments in bold and my answers in red.
Fig.2: Please explain the need for this figure. I could not find discussion on this data.
Thank you, we have added an explanation, and included a statement in the discussion highlighting the need to consider survivorship in other pathologies other than pancreatitis and pancreatic cancer .
Fig.3: Although most conclusions depend on Fig.3, there is no detailed explanation.
Thank you, I have added an extra line in the results section to expand on this figure, and a statement in the discussion for zinc, anaemia and Vitamin D to explain the impact of this further.
Table 3: Literature should be properly cited (please check the reference numbers).
Line 41: Ref. 26 gives no information on the zinc absorption mechanism.
Line 45: Ref.28 should be Ref.29 (Check the validity of references throughout the text).
Apologies, in formatting, the reference list had changed to a numerical list, and included the title “references” as number 1- shifting all the reference numbers up by 1, I have corrected this, which corrects the issues raised in these three points. Thank you for highlighting this.
Line 65: "high dose of zinc deficiency" should be checked.
Thank you, this has been amended to “high incidence of zinc deficiency”
Line 52-53: There appears to be no evidence in this study that high-dose oral zinc supplementation improves zinc levels.
Thank you for highlighting this, we debated this when designing the study, and we decided to only collect the annual blood test results. We have protocols for treating these deficiencies within our institution. Patients with deficiencies were treated and their levels reassessed after 3 months of treatment and if there was some but not sufficient improvement, a further 3-month course of treatment was completed, if there was no improvement, intravenous or intramuscular supplementation was provided. We did not include these results as there was no way to standardize or collate them (primary care repeated the blood tests, therefore the assays were different) and the time frames were very different. I have added a statement to that effect in the methodology, and a statement to expand on the outcomes in the results section (where we had commented on one patient needing intravenous iron) to include one patient who had intravenous fat-soluble vitamins and the resolution of deficiency with the others. We did not have sufficient data to carry out repeated measures assessments on individual patients (as we had hoped in planning), and I have added a statement to that effect in the limitations section of the discussion.
Thank you
Reviewer 2 Report
The manuscript “Do patients benefit from micronutrient supplementation fol- 2
lowing pancreatico-duodenectomy”. This study has implications for regular postoperative monitoring of mineral and vitamin levels in patients with Pancreatico-duodenectomy. However, the manuscript still need more improvement before can be accepted by the journal.
1. Pancreatico-duodenectomy affects the absorption of trace elements, what may be the mineral levels of Pancreatico-duodenectomy patients before supplementation with trace elements such as zinc, selenium, and copper? How soon it takes for supplementation to improve significantly. High-dose zinc supplementation may improve zinc deficiency levels, but has not reduced the risk of disease associated with zinc deficiency levels.
Pancreatico-duodenectomy will affect the absorption of trace elements, before supplementation of zinc, selenium, copper and other trace elements, what is the lower range of pancreatico-duodenectomy patients than normal patients? Patients have a reduced ability to absorb selenium, selenium supplementation may also face the problem of zinc supplementation, the safe dose range of selenium is very narrow, patients use high doses of selenium supplementation, will there still be selenium overdose problems? A large part of the patient's high dose of supplemented minerals and vitamins will be metabolized and excreted? Do unabsorbed high doses of minerals affect the structure of the intestinal flora?
2. Patients with pancreatico-duodenectomy have significantly reduced absorption of essential minerals and vitamins. Is the absorption of harmful substances, such as the heavy metal cadmium, also reduced? For smokers, will pancreatico-duodenectomy patients have lower cadmium levels than healthy people?
None
Author Response
Thank you for you comments, I have answered the queries raised below, with your initial comments in bold and our answers in red.
Pancreatico-duodenectomy affects the absorption of trace elements, what may be the mineral levels of Pancreatico-duodenectomy patients before supplementation with trace elements such as zinc, selenium, and copper? How soon it takes for supplementation to improve significantly. High-dose zinc supplementation may improve zinc deficiency levels, but has not reduced the risk of disease associated with zinc deficiency levels.
Thank you, the data we have included in on those who are not on high dose supplementation, but all patients are on standard supplements (multivitamin and mineral and a calcium and vitamin D). Unfortunately, we do not have sufficient data to comment on whether supplementation reduces the risk of disease associated with biochemical deficiencies. I have suggested further work includes the manifestations of micronutrients deficiency in the discussion, and have added a statement to reflect this.
Pancreatico-duodenectomy will affect the absorption of trace elements, before supplementation of zinc, selenium, copper and other trace elements, what is the lower range of pancreatico-duodenectomy patients than normal patients?
Apologies, I am not sure I can answer this, we did not have a control group within this study, and I have discussed this in the limitations section of our discussion.
Patients have a reduced ability to absorb selenium, selenium supplementation may also face the problem of zinc supplementation, the safe dose range of selenium is very narrow, patients use high doses of selenium supplementation, will there still be selenium overdose problems?
Thank you, toxicity is a concern, we did not observe any cases of toxicity, but we were not providing very high levels of supplementation. We provided 200ug of selenium to treat our selenium deficiencies (included in the methodology), this is well below the toxic level of selenium dosing (the maximum safe dose in the UK is proposed at 450ug. (https://assets.publishing.service.gov.uk/government/uploads/system/uploads/attachment_data/file/339431/SACN_Selenium_and_Health_2013.pdf)
A large part of the patient's high dose of supplemented minerals and vitamins will be metabolized and excreted? Do unabsorbed high doses of minerals affect the structure of the intestinal flora?
This is an interesting question, unfortunately we did not assess the gut flora within this study and whilst I am aware of several studies asking the same question (i.e., Barra et al, J Endocrinol 2021; 250(2)R1-21, Calder et al, Adv Nutr 2022;13(5):S1-26), I did not find any studies within this cohort. I suspect that it will be difficult to differentiate any impact of a single intervention such as micronutrients in this patient cohort as there are many other factors that will impact the microbiome such as the impact of long term proton pump inhibitor use, asynchrony of delivery of bile and pancreatic bicarbonate to the small bowel, the presence of a blind loop of bowel, high incidence of diabetes, long term impacts of chemotherapy / radiotherapy etc., I have not added anything to the paper regarding this, as I feel it is outside of the scope of this paper, but I agree, a fascinating area for future research.
Patients with pancreatico-duodenectomy have significantly reduced absorption of essential minerals and vitamins. Is the absorption of harmful substances, such as the heavy metal cadmium, also reduced? For smokers, will pancreatico-duodenectomy patients have lower cadmium levels than healthy people?
This is a great point, I did not find any papers exploring cadmium levels in patients following pancreatico-duodenectomy, we do not measure cadmium, and my understanding is that uptake from the intestine is relatively poor in the general population the incidence of toxicity in the UK is low, and usually related to exposure to industrial airborne pollutants, rather than dietary sources. I understand the hypothesis that if a smoker had a lower baseline cadmium due to dietary malabsorption, they may have a lower cumulative cadmium level, compared to someone who had not had a pancreatico-duodenectomy. Interestingly, I did find a study in rats demonstrating the cadmium uptake was increased in the presence of iron depletion (Park et al, Toxicological Sciences 2002;68(2);288-294. In respect to this paper I have not added anything about cadmium, but I have added a statement to the effect that further work could include exploration of the impact of pancreatico-duodenectomy on other minerals not routinely assessed in clinical practice, as I agree this could be explored, and patients who have had intestinal surgery for other conditions, including bariatric surgery (who tend to be younger and have better long term survival), could be included.